# ROYAL SOCIETY
# OPEN SCIENCE

nanotechnology

ferric hydroxide, regeneration, bamboo nanocellulose, adsorption

**Author for correspondence:**
Ying Chen
e-mail: cylm@163.com

# Preparation and regeneration of iron-modified nanofibres for low-concentration phosphorus-containing wastewater treatment

Ying Luo[1], Min Liu[1,2], Ying Chen[1,2], Tingting Wang[1] and Wei Zhang[3]

[1]College of Architecture and Environment, Sichuan University, Chengdu 610065, People's Republic of China
[2]Sino-German Centre for Water and Health Research, Chengdu 610065, People's Republic of China
[3]State Key Laboratory of Polymer Materials Engineering, Polymer Research Institute at Sichuan University, Chengdu 610065, People's Republic of China

YC, 0000-0002-8388-5267

In this study, nanocellulose (CNFs) was prepared by a mechanical shearing method, a simple and pollution-free process. Iron hydroxide was loaded on nanocellulose, a natural macromolecule derived from bamboo, to produce the second-generation iron-loaded nanocellulose for the removal of low-concentration phosphorus from wastewater. We found that the best modified ferric salt was ferric chloride. When the mass ratio of $Fe(OH)_3$ and CNFs was 1.5 : 1, freeze-drying with liquid nitrogen yielded the best adsorption performance. The adsorption process of $Fe(OH)_3$@CNFs followed the pseudo-second-order kinetics and belonged to chemical adsorption. Regeneration experiments showed that after 10 cycles of adsorption–regenerations of the adsorbent, the phosphorus adsorption efficiency was still stable at 80% of the initial material. The prepared adsorbent was characterized by the BET surface area measurement, scanning electron microscopy and FT-IR. The surface morphology, pore size and elements of materials before and after iron loading were analysed. Compared with other adsorbents, the phosphorus removal performances of the second-generation iron-loaded nanocellulose were superior. Compared with the first-generation material, the second-generation adsorbent is simpler and more environmentally friendly.

# 1. Introduction

Water pollution such as eutrophication is a worldwide problem [1]. Controlling phosphorus levels is an important means to abate water eutrophication and improve the water environment. China's 'Discharge standard of pollutants for municipal wastewater treatment plant' (GB18918-2002) stipulates that the first-class A standard of total phosphorus emission concentration is $0.5 \, \mathrm{mg \, l^{-1}}$. In recent years, some local standards have been promulgated to make the total phosphorus emission standards more stringent, even down to $0.3 \, \mathrm{mg \, l^{-1}}$, reaching the surface water category IV standard of China. Thus, there is an urgent need to develop efficient and stable technologies to remove low-concentration phosphorus from wastewater.

Adsorption is commonly used for treating low-concentration pollutants [2,3], due to the facile operation and low energy consumption [4–6]. In the removal of phosphorus, many natural adsorbents are developed, including fly ash [7–9], zeolite [10–14] and diatomite [15,16]. Fly ash has a strong adsorption capacity but may pollute the environment [17]. Zeolites have a large specific surface area and strong electrostatic attraction. However, natural zeolite exhibits a poor performance in removing anions directly from water. Diatomite has a large specific surface area and strong adsorption capacity for many pollutants. But due to the negative surface charges, diatomite is often used to adsorb positively charged heavy metal ions.

Cellulose is the most abundant and reproducible natural macromolecular material on Earth, and it possesses some adsorptive properties [18] and can be used as an adsorbent [19]. Bamboo is a cellulose-rich material with a high cellulose content [20]. China has a large bamboo production, and bamboo grows fast. The bamboo resources can be quickly regenerated. It is a high-quality cellulose material. In recent years, due to the high surface area of nanomaterials, researchers have been working on the preparation of nanomaterials as nanosorbents [21–29]. And there are few studies on bamboo-based cellulose at the nanoscale. Therefore, it is valuable to study the application of bamboo nanofibres adsorbent.

Natural bamboo materials have a weak adsorption capacity. It is generally necessary to modify the material to improve its adsorption performance. A large number of studies have shown that metal oxides and metal hydroxides, such as those of Fe [30–32], Al [33,34], Mn [35], Mg [2] and Zr [36,37], have a good adsorption activity on phosphorus. Among them, Fe and Al are the two most studied metals. Rebosura *et al.* [38] have shown that iron salts are commonly used in wastewater treatment and can also remove phosphorus from wastewater. At present, the development of highly efficient, environmentally friendly and repeatedly usable adsorbents is a research hotspot for phosphorus removal. In particular, the regeneration of adsorbent has attracted growing attention from researchers. Regeneration is the reverse process of adsorption. Reagent regeneration allows the recycling of the adsorbent, prolonging the life cycle of the adsorbent and reducing the cost of treatment.

Our research group has developed the first-generation iron-loaded nanocellulose ($Fe(OH)_3$@CNFs) for phosphorus removal. Cui *et al.* [39] verified it can remove phosphorus from wastewater. But, cellulose produced by TEMPO is expensive and limits the mass production or applications. Therefore, in this study, the pristine bamboo pulp without drying or heating treatment was directly used as raw material to prepare nanocellulose using mechanical shearing. Iron hydroxide was loaded on this bamboo-based nanocellulose, developing the second-generation iron-loaded nanocellulose to improve the removal of the low-concentration phosphorus from wastewater. The effects of different iron salts, $Fe(OH)_3$ and nanocellulose (CNFs) mass ratios, freezing concentration and freezing mode modification on the second-generation iron-loaded nanocellulose adsorption performance were studied. The regeneration properties of the material were also evaluated.

# 2. Material and methods

## 2.1. Chemical reagents

The chemical reagents mainly included: sodium hydroxide (NaOH), potassium phosphate monobasic ($KH_2PO_4$), sulfuric acid ($H_2SO_4$), hydrogen chloride (HCl), ascorbic acid ($C_6H_8O_6$), ammonium molybdate (($NH_4)_6MO_7O_4 \cdot 4H_2O$), potassium antimony(III)oxide tartrate hemi-hydrate ($C_4H_4KO_7Sb \cdot 1/2H_2O$), ferric nitrate ($Fe(NO)_3 \cdot 9H_2O$), ferric chloride ($FeCl_3 \cdot 6H_2O$), sodium chloride (NaCl), sodium nitrate ($NaNO_3$), sodium sulfate ($Na_2SO_4$), ammonium chloride ($NH_4Cl$), sodium carbonate ($Na_2CO_3$), potassium persulfate ($K_2S_2O_8$), ferric sulfate ($Fe_2(SO_4)_3$) and ferric reagent (LH-FE). Ferric sulfate was

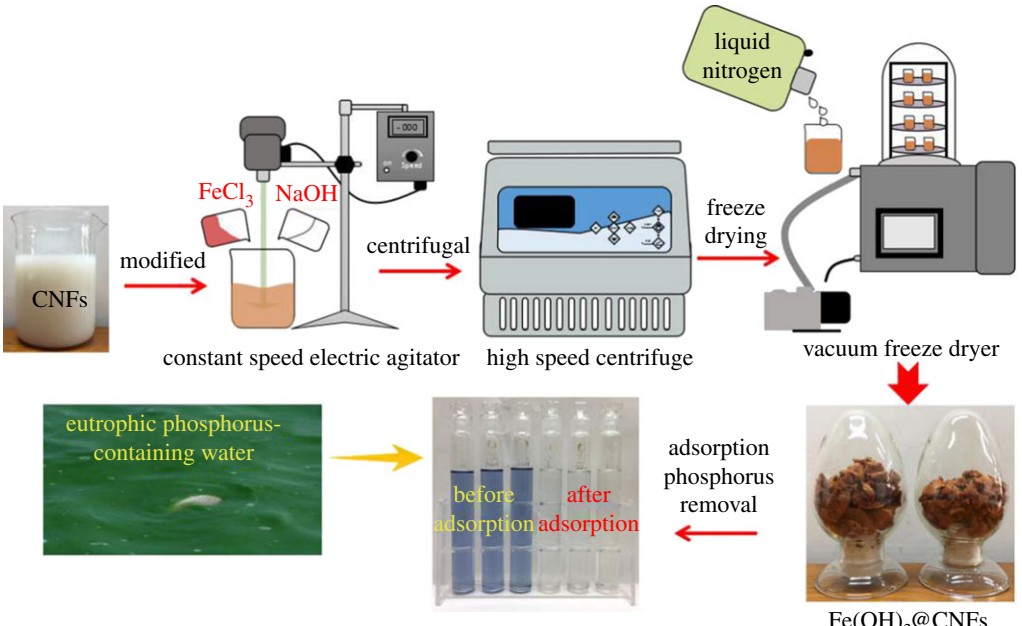

**Figure 1.** Schematic diagram of the optimized materials experimental programme.

purchased from Xilong Chemical Co., Ltd, China, and LH-FE reagent was purchased from Beijing Lianhua Yongxing Technology Development Co., Ltd. The other reagents were purchased from Kelong Chemical Reagent Factory in Chengdu. All the chemicals were analytical grade. The simulated phosphorus-containing wastewater was prepared using potassium dihydrogen phosphate. The bamboo CNFs used in the experiment were provided by the State Key Laboratory of Polymer Materials Engineering, Polymer Research Institute at Sichuan University, China, in a 1 wt% solution.

## 2.2. Preparation of the second-generation iron-loaded nanocellulose and its phosphorus adsorption experiments

The experimental procedure is shown in figure 1. The nanocellulose suspension was prepared by mechanical shearing of the pristine bamboo pulp as raw feedstock material [40]. The ferric chloride solution and sodium hydroxide solution were properly dosed into the CNF suspension. After stirring at 25°C for 10 h, the solution was washed by centrifugation, and the centrifuged solid was freeze-dried.

The experimental procedure consisted of adding 100 ml of the simulated wastewater containing 2 mg l$^{-1}$ phosphate to 250 ml Erlenmeyer flask. After adding 10 mg adsorbent, the mixture was shaken in a constant temperature oscillation box at 25°C at 150 r.p.m. for 24 h. The suspension was filtered with a 0.45 µm polyethersulfone filter. Then the phosphorus content in the solution was measured before and after adsorption. When the impact of pH was explored, pH was adjusted to 4, 7 and 10. In other experiments, the pH was adjusted to 7. Three parallel experiments were performed for each group.

## 2.3. Regeneration experiments

The collected adsorbent was washed six times with deionized water and dried at 60°C for 24 h. In the sixth cycle of regeneration, the drying temperature of the adsorbent was raised to 105°C. The NaOH solution served as a desorption solution. The dried adsorbent was added to the desorption solution, shaken and immersed for 24 h in a 25°C constant temperature oscillator.

## 2.4. Analytical methods

The ammonium molybdate spectrophotometric method was employed to determine the phosphorus concentration in the solution. The iron concentration in the solution was measured using the bathophenanthroline spectrophotometric method. The mass concentration of blank nanofibres was determined by the dry weight method. Surface morphology and element content were analysed by a

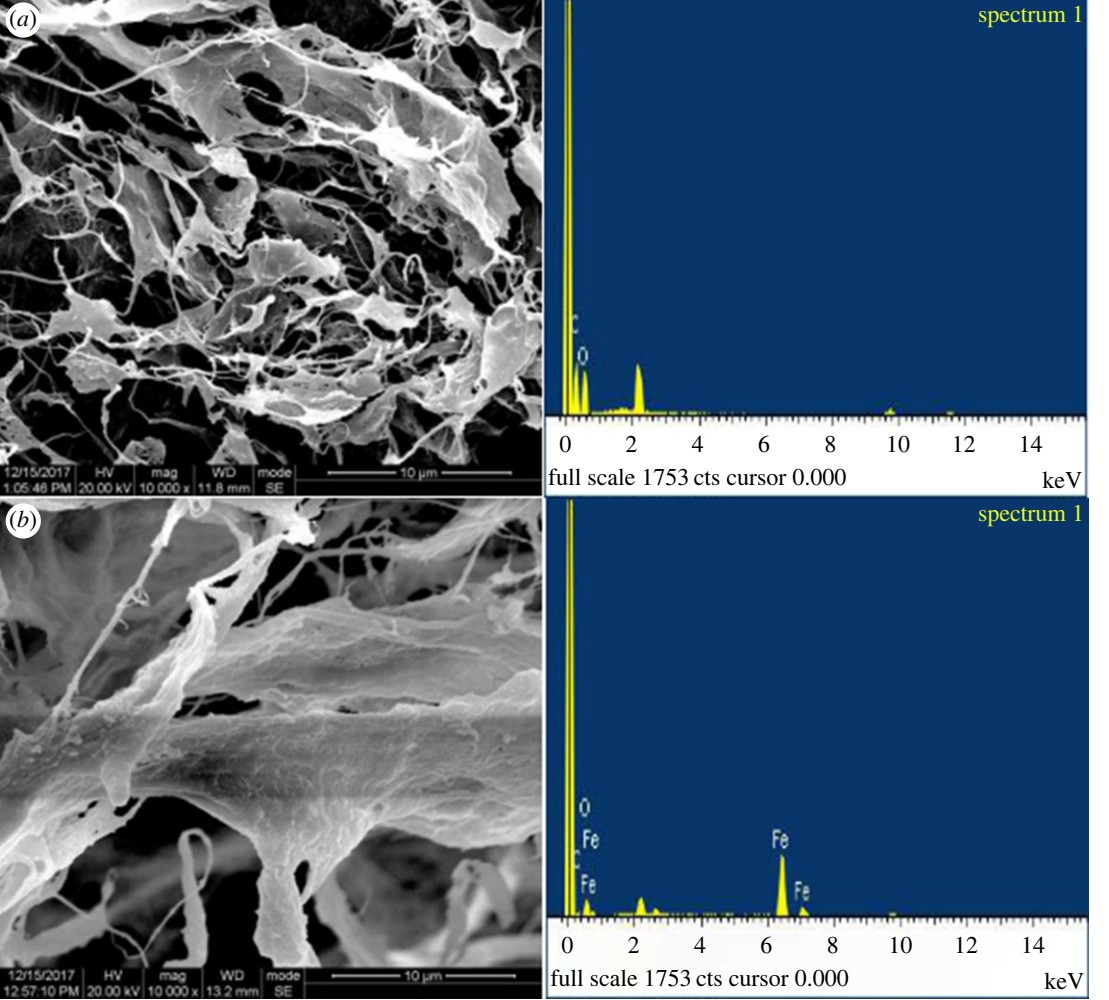

**Figure 2.** Scanning electron microscope (*a*) CNFs × 10 000 and (*b*) Fe(OH)₃@CNFs × 10 000.

LEO 1530 scanning electron microscope (Zeiss, Germany). FT-IR analysis was performed using a Nicolet 6700 Fourier transform infrared spectrometer. The equilibrium adsorption capacity is calculated as shown in the following formula:

$$q_e = \frac{(C_0 - C_e)V}{m},\qquad(2.1)$$

where $C_0$ is the initial concentration (mg l$^{-1}$), $C_e$ is the concentration of the solution after adsorption (mg l$^{-1}$), $q_e$ is the adsorption capacity at adsorption equilibrium (mg g$^{-1}$), $V$ is the volume of desorption solution (l) and $m$ is the mass of the adsorbent (g).

# 3. Results and discussions

## 3.1. Characterization

The amount of phosphorus adsorbed on this second-generation iron-loaded nanocellulose increased significantly compared with CNFs without the iron oxide modification. To better analyse the modified material, the material prepared under optimal conditions was characterized.

### 3.1.1. SEM and EDS analyses

Figure 2 shows the fibrous structures of the pristine CNFs and the iron-loaded CNFs. The surface deposition of Fe(OH)₃ on nanocellulose is confirmed by the EDS analysis. Table 1 shows that the surface-attached iron on CNFs reached a weight percentage of 33.65%.

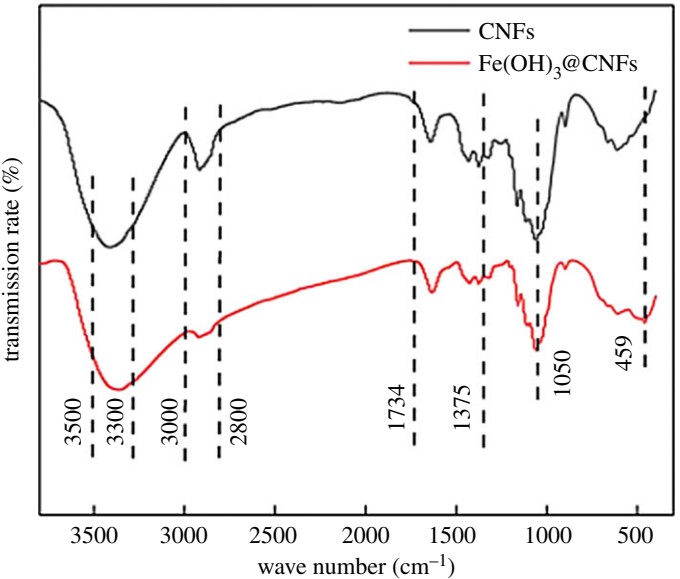

**Figure 3.** Infrared spectra of CNFs before and after modification.

**Table 1.** Analysis of CNFs elements before and after modification.

| elements | CNFs (%) | Fe(OH)$_3$@CNFs (%) |
| --- | --- | --- |
| C | 55.1 | 53.81 |
| O | 44.9 | 12.54 |
| M | 0 | 33.65 |

**Table 2.** Analysis of specific surface area and pore size of CNFs before and after modification.

| analyte | $S_{BET}$ (m$^2$ g$^{-1}$) | Langmuir specific surface area (m$^2$ g$^{-1}$) | total pore volume (cm$^3$ g$^{-1}$) | average pore diameter (nm) | BJH average pore diameter (nm) |
| --- | --- | --- | --- | --- | --- |
| CNFs | 14.481 | 19.412 | 0.353 | 97.379 | 200.540 |
| Fe(OH)$_3$@CNFs | 138.907 | 189.214 | 0.122 | 3.518 | 2.052 |

### 3.1.2. Specific surface area and pore size analysis

Specific surface area, pore size distribution, pore size and pore volume are important for the adsorption performance of adsorbents. Table 2 shows that the specific surface area increased from 14.48 to 138.91 m$^2$ g$^{-1}$ after surface modification with Fe(OH)$_3$. The specific surface area of the iron-loaded CNFs is about 9.6 times that of the pristine CNFs. The total pore volume reduced from 0.35 to 0.12 cm$^3$ g$^{-1}$ after the iron loading. The average pore diameter also reduced from 97.38 to 3.52 nm after modification. Fe(OH)$_3$ is loaded on the surface of the material and inside the pores after the material modification, the particles of ferric hydroxide may increase the specific surface area of the material. The reduction in pore volume and pore diameter is presumably because of the surface deposition of Fe(OH)$_3$.

### 3.1.3. FT-IR analysis

Figure 3 compares the IR spectra of the pristine CNFs and the iron-loaded CNFs, which have similar patterns of surface functional groups. According to the relevant literature [41–43], the strong wide peaks between 3300 and 3500 cm$^{-1}$ were –OH stretching vibration peaks. The peak near 2800–3000 cm$^{-1}$ corresponded to the C–H symmetric stretching vibration absorption peak. The peak

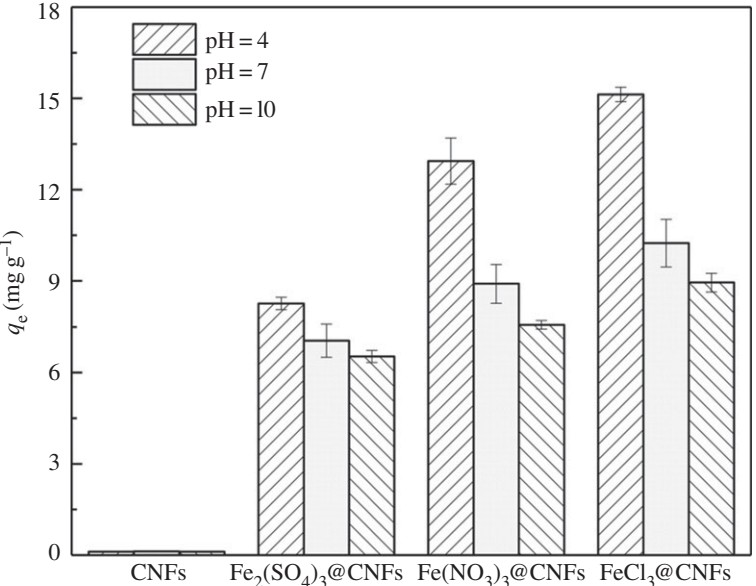

**Figure 4.** Phosphorus removal effect of CNFs and Fe(OH)$_3$@CNFs under different pH conditions.

between 1375 and 1734 cm$^{-1}$ and the peak near 899 cm$^{-1}$ corresponded to the C–H deformation vibration absorption peak. The peak near 1050 cm$^{-1}$ corresponded to C–O vibration. Compared with the pristine CNFs, the second-generation iron-loaded nanocellulose appeared to have a new peak at 459 cm$^{-1}$, which could be ascribed to a functional group (Fe-OH) or the iron-hydroxyl bond.

## 3.2. Comparison of adsorption capacity of phosphorus

The adsorption efficiency of phosphorus on the pristine and the iron-loaded CNFs were studied at pH 4, 7 and 10. Three kinds of iron-loaded CNFs were prepared using ferric chloride, ferric sulfate and ferric nitrate. The modified CNFs had the same mass ratio of Fe(OH)$_3$ and CNFs of 1 : 1. As shown in figure 4, the pristine CNFs had almost negligible phosphorus removal. By contrast, the phosphorus adsorption of the three modified CNFs was significantly improved and ferric chloride had the best phosphorus removal performance, but exhibited consistent pH dependence, which agrees with the previous study [44]. Under similar experimental conditions [45–48], the second-generation iron-loaded nanocellulose had better adsorption effect for phosphorus than most other adsorbents. The modification method is easy to operate. The phosphorus removal capacity of the optimized adsorbent was significantly enhanced.

## 3.3. Optimization of the preparation conditions for the second-generation iron-loaded nanocellulose

### 3.3.1. Effect of the freezing concentration and freezing method

To investigate the effects of the second-generation iron-loaded nanocellulose prepared by varying the freezing concentrations and freezing methods on phosphorus removal, the prepared material solution was separately dispersed in water to 0.1, 1, 3, 5 and 10 wt%. The solutions were frozen with liquid nitrogen and a refrigerator. After freezing, the solutions were dried in a freeze dryer.

According to figure 5, the adsorption capacity of the adsorbent slightly decreased after the freezing treatment by liquid nitrogen when the concentrations of the iron-loaded nanocellulose increased from 0.1 to 10 wt%. However, under the refrigerator freezing treatment, the adsorption capacity remained almost unchanged. The difference in the adsorption capacity after liquid nitrogen freeze-drying or freeze-drying in the refrigerator may be caused by the morphology changes of adsorbents during freezing/drying processes. Liquid nitrogen freezes the adsorbents quickly, and the water in the adsorbent solution can quickly become small, dense ice crystals. After vacuum drying, as the water is removed or vaporized, the adsorbent developed a porous structure with high specific surface area [49,50]. By contrast, the

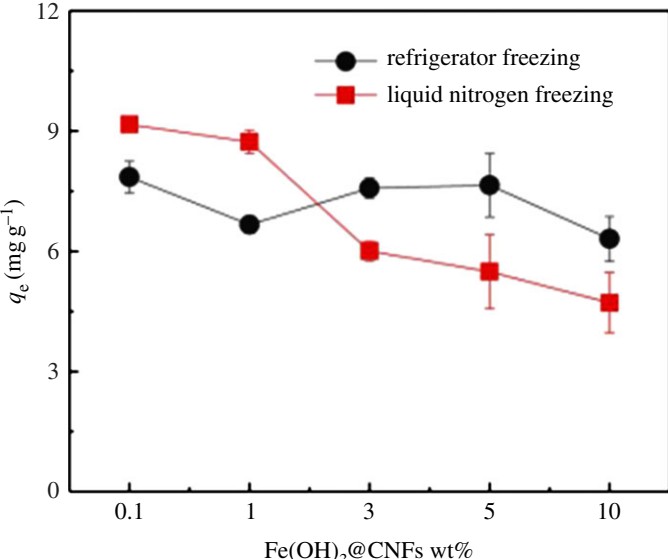

**Figure 5.** Phosphorus removal effect of different concentrations of Fe(OH)$_3$@CNFs after freezing in refrigerator and liquid nitrogen.

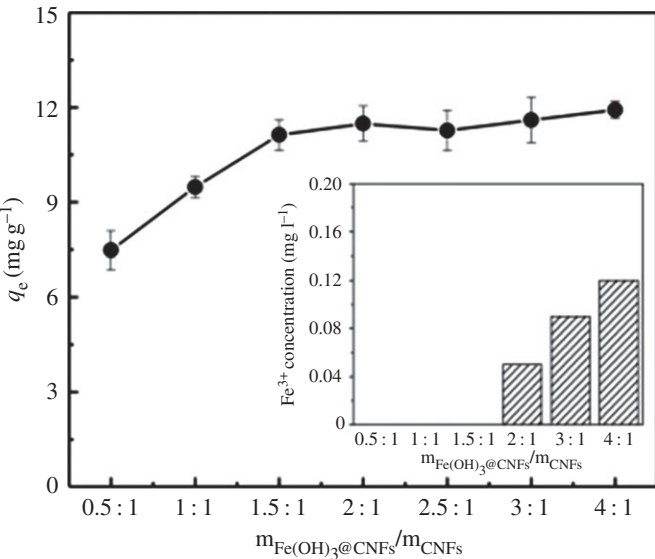

**Figure 6.** Phosphorus removal effect of the second-generation iron-loaded nanocellulose. Modified by different Fe(OH)$_3$ and CNFs mass ratio. The inset shows iron dissolution.

freezing time in the refrigerator is longer; accordingly, the ice crystals formation took a longer time during the freezing process and the ice crystals formed are larger. Therefore, liquid nitrogen freeze-dried materials were superior to refrigerator freeze-drying. At high mass concentrations, the freezing time of the adsorbents may be longer, which reduces the formation of the porous structures of the adsorbents. Unless indicated, the iron-loaded adsorbents were all treated under liquid nitrogen freezing at 1 wt%.

### 3.3.2. Effect of the mass ratio of Fe(OH)$_3$ and CNFs

To determine the optimal fabrication condition, the CNFs were modified by varying the mass ratio of Fe(OH)$_3$/CNFs from 0.5 : 1 to 4 : 1 and the adsorption capacities were evaluated. Figure 6 shows that when the Fe(OH)$_3$/CNF ratio was 0.5 : 1 ∼ 1.5 : 1, the adsorption capacity almost linearly increased with the mass ratio. When the Fe(OH)$_3$/CNF ratio was 2 : 1 ∼ 4 : 1, the increase in adsorption capacity became less significant, indicating that the maximum active adsorption sites on iron-loaded CNFs could be reached.

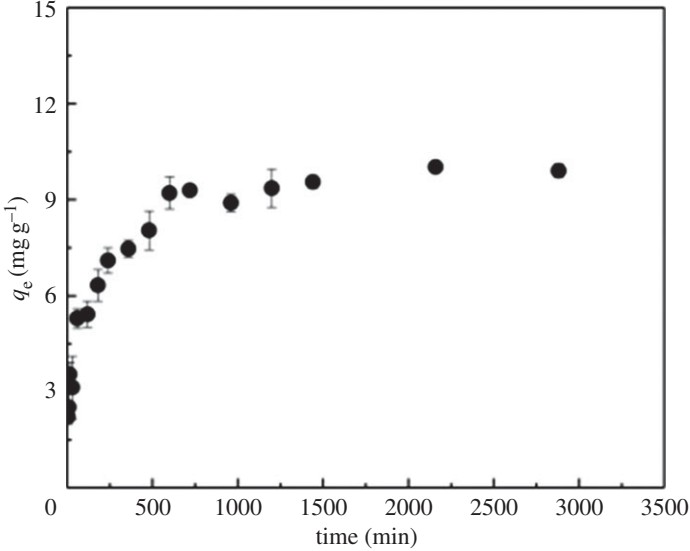

**Figure 7.** Sorption kinetics of phosphate on Fe(OH)$_3$@CNFs.

**Table 3.** Parameters for the adsorption kinetics model of Fe(OH)$_3$@CNFs.

| adsorbent | pseudo-first-order model | | | pseudo-second-order model | | | Morris–Weber intraparticle diffusion model | |
| | $q_e$ | $k_1$ | | $q_e$ | $k_2$ | | $k_{int}$ | |
| | mg g$^{-1}$ | min$^{-1}$ | $R^2$ | mg g$^{-1}$ | g (mg min)$^{-1}$ | $R^2$ | g (mg min$^{1/2}$)$^{-1}$ | $R^2$ |
| Fe(OH)$_3$@ CNFs | 8.95 | 0.011 | 0.809 | 10.14 | 0.001 | 0.998 | 0.158 | 0.816 |

Before the mass ratio of Fe(OH)$_3$/CNFs was 1.5 : 1, no iron ions overflowed in the solution. With a 2 : 1 ratio, iron ions overflowed. The iron ions concentration increased as the mass ratio increased. CNFs could fully bind to Fe(OH)$_3$ when the mass ratio of Fe(OH)$_3$/CNFs was 1.5 : 1. With the increase in the Fe(OH)$_3$, some Fe(OH)$_3$ was not tightly bound to CNFs, which could be easily shaken off during the oscillating adsorption process. According to the phosphorus adsorption capacity and the amount of iron ion overflowed, and the mass ratio of Fe(OH)$_3$/CNFs for modification was 1.5 : 1.

After optimization, the second-generation iron-loaded nanocellulose is used to adsorb phosphorus solution with an initial concentration of 2 mg l$^{-1}$, and the adsorption capacity is 11.45 mg g$^{-1}$. Cui et al. [39] developed the first-generation Fe(OH)$_3$@CNFs. Cui et al. used Fe(OH)$_3$@CNFs to adsorb phosphorus, and the maximum adsorption capacity was 142.86 mg g$^{-1}$. The data in this study is far lower than that of Cui et al., mainly because of the different initial phosphorus concentrations (2 vs 500 mg l$^{-1}$). However, when tested under the same condition (500 mg l$^{-1}$), the maximum adsorption capacity of the present iron-loaded nanocellulose is 152.11 mg g$^{-1}$, which is superior to that of Cui et al.

## 3.4. Adsorption kinetics

The adsorption kinetics of phosphate on Fe(OH)$_3$@CNFs increased rapidly in the first 11 h, and then reached equilibrium as shown in figure 7. The pseudo-first-order, the pseudo-second-order and Morris–Weber internal diffusion models were used to fit the adsorption kinetic data with the major fitting parameters shown in table 3. The pseudo-second-order kinetic model yielded a higher correlation coefficient ($R^2$) of 0.998 than the pseudo-first-order kinetic model, suggesting that the adsorption process could be chemical adsorption. After fitting the kinetic data with Morris–Weber intraparticle diffusion model, indicating that internal diffusion was not the only step to control the adsorption process. The adsorption process was divided into two processes: surface adsorption and internal diffusion.

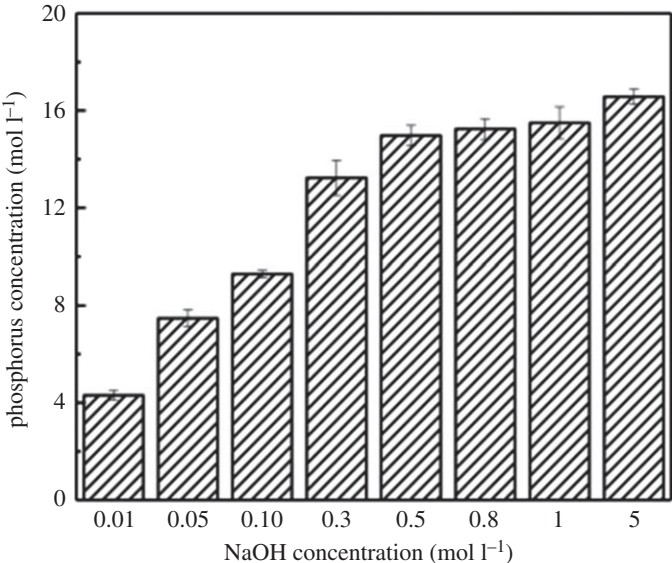

**Figure 8.** Effect of NaOH concentration on desorption.

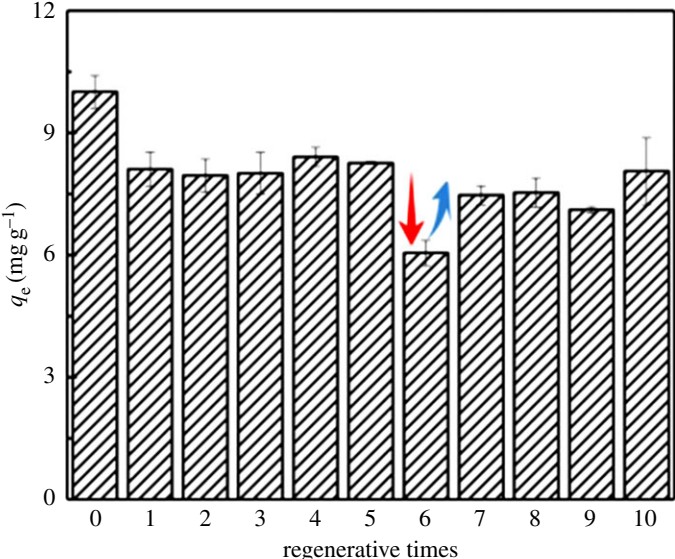

**Figure 9.** Effect of regeneration times on adsorption efficiency of the second-generation iron-modified nanocellulose.

## 3.5. Regeneration studies

### 3.5.1. Effect of NaOH solution concentration

Studies have shown that the sodium hydroxide (NaOH) cleaning is effective for phosphorus desorption and adsorbent regeneration and can be re-used many times after regeneration [51,52]. The NaOH solutions with different concentrations of $0.01-5\,mol\,l^{-1}$ were used to remove phosphate from used adsorbents. Figure 8 shows that with the increase in NaOH concentration, the desorbed amounts of phosphorus from the adsorbent also increased and reached a plateau at an NaOH concentration of $0.5\,mol\,l^{-1}$. Therefore, $0.5\,mol\,l^{-1}$ NaOH solution was selected as the desorption solution.

### 3.5.2. Effect of the regeneration times on the adsorption–desorption of phosphorus on the second-generation iron-loaded nanocellulose

After 10 cycles of adsorption and desorption using the above regeneration process, the adsorption capacity slightly decreased by 19% compared to the initial adsorption capacity as shown in figure 9.

In the sixth cycle of regeneration, the drying temperature of the adsorbent was raised to 105°C, which led to the decrease in phosphorus adsorption. In the seventh cycle of regeneration, the drying temperature was reduced to 60°C, which increased the adsorption capacity to more than 80% of the initial level. Clearly, the high drying temperature caused the decline of adsorption affinity probably because of the reduction in adsorption sites. However, lowering the drying temperature could recover the adsorption capacity of the iron-loaded CNFs.

# 4. Conclusion

In this study, nanocellulose was produced using mechanical shearing of wet bamboo pulp as raw material. This method does not use TEMPO reagent, which simplifies the synthesis procedure and reduces secondary pollution from adsorbent production. After optimizing the preparation conditions of the iron-loaded nanocellulose, ferric chloride was determined to be the best modifier chemical. The specific surface area of the modified adsorbent increased from 14.48 to 138.91 $m^2\,g^{-1}$. When the mass ratio of $Fe(OH)_3$ to CNFs was 1.5 : 1, the addition of $FeCl_3$ solution and freeze-drying with liquid nitrogen yielded the highest adsorption capacity on phosphorus. The adsorption process of $Fe(OH)_3@CNFs$ conforms to the pseudo-second-order kinetic model, suggesting that the adsorption process is chemical adsorption. After 10 desorption–adsorption regeneration cycles, the phosphorus adsorption capacity remained stable. The second-generation iron-loaded nanocellulose can be re-used at least 10 times. This iron-loaded nanocellulose holds great promise as highly effective and economically viable phosphorus adsorbent.

Data accessibility. Data available from the Dryad Digital Repository: https://doi.org/10.5061/dryad.c53m9d5 [53].

Authors' contributions. Y.L. mainly wrote the manuscript and conducted the major experiments; M.L. and Y.C. conceived the research ideas and experimental plans and evaluated data and the manuscript; T.W. participated in the experiments; W.Z. provided raw materials and participated in manuscript writing and revision.

Competing interests. We have no competing interests.

Funding. This work was supported by the Science and Technology Major Projects of Sichuan Province 'Technology Integration and Demonstration of Stability and Standard Achievement in Urban Sewage Treatment Plant' (grant no. 2019YFS0501), and the Fundamental Research Funds for the Central Universities 'Application of Modified Nanocellulose in Wastewater Treatment'.

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
