## [Reviewer comments · Royal Society Open Science]

Review History

RSOS-190764.R0 (Original submission)

Review form: Reviewer 1

Is the manuscript scientifically sound in its present form?

Yes

Are the interpretations and conclusions justified by the results?

Yes

Is the language acceptable?

No

Do you have any ethical concerns with this paper?

No

Have you any concerns about statistical analyses in this paper?

No

Recommendation?

Accept with minor revision (please list in comments)

Comments to the Author(s)

Dear respected editor

Thank you very much for letting me the opportunity to reviews some scientific papers in your valuable journal.

Regarding the manuscript under consideration, (entitled: Preparation and regeneration of iron-modified nanofibres for low concentration phosphorus-containing wastewater treatment), the manuscript and the obtained results are interesting and well organized. I suggest this manuscript would be suitable for publication in your respected journal after minor revision. My comments are:

1: Following references must be included in the introduction

J. Inorg. Organomet. Polym. 29 (2019) 346-358

Compos. Part B. 166 (2019) 382-400.

Microchem. J. 145 (2019) 18-25.

J. Inorg. Organomet. Polym. 29 (2019) 229-247.

J. Mol. Liq. 253 (2018) 72-82.

J. Mol. Liq. 242 (2017) 364-374.

J. Mol. Liq. 248 (2017) 302-313.

J. Mol. Struct. 1143 (2017) 462-471.

J. Mol. Struct. 1086 (2015) 223-231.

J. Inorg. Organomet. Polym. (2017) 1-14.

Spectrochimica Acta Part A: Molecular and Biomolecular Spectroscopy 222(2019)117195

Adv. Powder Technol., 23 (2012) 757-760.

2: There are some grammatical errors in the manuscript. Hence, the manuscript should be carefully checked and necessary corrections should be done.

3: In the experimental part (2.1 Chemical reagents), the chemical structure of the used chemicals must be written.

Review form: Reviewer 2

Is the manuscript scientifically sound in its present form?

Yes

Are the interpretations and conclusions justified by the results?

Yes

Is the language acceptable?

Yes

Do you have any ethical concerns with this paper?

Yes

Have you any concerns about statistical analyses in this paper?

Yes

Recommendation?

Major revision is needed (please make suggestions in comments)

Comments to the Author(s)

The manuscript reports on Preparation and regeneration of iron-modified nanofibres for low concentration phosphorus-containing wastewater treatment. Authors are attempted to develop iron-modified nanofibres for the treatment for low concentration phosphorus-containing wastewater treatment. It needs major revision before accepting for publication.

In introduction section lack of literature review is missing, authors should read below relevant reference and consider for citation.

Nanophotocatalysis and Environmental Applications, 139-169, 2020.

International Journal of Hydrogen Energy, doi.org/10.1016/j.ijhydene.2019.03.149. 2019

Nanophotocatalysis and Environmental Applications, 83-105, 2019

Polyhedron 120, 169-174, 2016.

Proper scale is missing in SEM images, better to provide details in the image.

The Conclusion section is too short, it should be written properly. And also mention like can be reused many times, it is better to mention how many times can be used.

English and grammatical errors should be rectified during the revision of the paper.

Review form: Reviewer 3

Is the manuscript scientifically sound in its present form?

Yes

Are the interpretations and conclusions justified by the results?

Yes

Is the language acceptable?

Yes

Do you have any ethical concerns with this paper?

Yes

Have you any concerns about statistical analyses in this paper?

No

Recommendation?

Accept with minor revision (please list in comments)

Comments to the Author(s)

How to improve the economy and recycling degree of low concentration phosphorus wastewater treatment is one of the difficulties of municipal domestic wastewater treatment. This paper described the adsorption of low concentration phosphorus from simulated waste water source using bamboo-based nanocellulose fiber loaded with iron hydroxide (Fe(OH)₃@CNFs). As an environmentally friendly adsorbent material, it has great potential for application. On the other hand, it can provide technical support for China's continuous improvement of sewage discharge standards, and it can also provide a feasible way for the recovery and utilization of phosphorus resources.

1. There are different kinds cellulose materials in nature, the authors should tell what is the advantage of bamboo compared with others.
2. Loading of metal/metal oxides to carrier material result in decreased surface area and porosity due to void volume blockages. However, what we observed here is a significant increase in surface area after loading of $\text{Fe}(\text{OH})_3$?
3. Figures 4 and 5 are similar in meaning and suggest integration.
4. Supplementary dynamics research to enhance the integrity of the article.

Decision letter (RSOS-190764.R0)

08-Jul-2019

Dear Professor Chen:

Title: Preparation and regeneration of iron-modified nanofibres for low concentration phosphorus-containing wastewater treatment
Manuscript ID: RSOS-190764

The editor assigned to your manuscript has now received comments from reviewers. We would like you to revise your paper in accordance with the referee and Subject Editor suggestions which can be found below (not including confidential reports to the Editor). Please note this decision does not guarantee eventual acceptance.

Please submit your revised paper before 31-Jul-2019. Please note that the revision deadline will expire at 00.00am on this date. If we do not hear from you within this time then it will be assumed that the paper has been withdrawn. In exceptional circumstances, extensions may be possible if agreed with the Editorial Office in advance. We do not allow multiple rounds of revision so we urge you to make every effort to fully address all of the comments at this stage. If deemed necessary by the Editors, your manuscript will be sent back to one or more of the original reviewers for assessment. If the original reviewers are not available we may invite new reviewers.

RSC Associate Editor:
Comments to the Author:
(There are no comments.)

RSC Subject Editor:
Comments to the Author:
(There are no comments.)

Reviewers' Comments to Author:
Reviewer: 1

Comments to the Author(s)
Dear respected editor

Thank you very much for letting me the opportunity to reviews some scientific papers in your valuable journal.

Regarding the manuscript under consideration, (entitled: Preparation and regeneration of iron-modified nanofibres for low concentration phosphorus-containing wastewater treatment), the manuscript and the obtained results are interesting and well organized. I suggest this manuscript would be suitable for publication in your respected journal after minor revision. My comments are:

1: Following references must be included in the introduction

J. Inorg. Organomet. Polym. 29 (2019) 346–358

Compos. Part B. 166 (2019) 382–400.

Microchem. J. 145 (2019) 18–25.

J. Inorg. Organomet. Polym. 29 (2019) 229–247.

J. Mol. Liq. 253 (2018) 72–82.

J. Mol. Liq. 242 (2017) 364–374.

J. Mol. Liq. 248 (2017) 302–313.

J. Mol. Struct. 1143 (2017) 462–471.

J. Mol. Struct. 1086 (2015) 223–231.

J. Inorg. Organomet. Polym. (2017) 1–14.

Spectrochimica Acta Part A: Molecular and Biomolecular Spectroscopy 222(2019)117195

Adv. Powder Technol., 23 (2012) 757-760.

2: There are some grammatical errors in the manuscript. Hence, the manuscript should be carefully checked and necessary corrections should be done.

3: In the experimental part (2.1 Chemical reagents), the chemical structure of the used chemicals must be written.

Reviewer: 2

Comments to the Author(s)

The manuscript reports on Preparation and regeneration of iron-modified nanofibres for low concentration phosphorus-containing wastewater treatment. Authors are attempted to develop iron-modified nanofibres for the treatment for low concentration phosphorus-containing wastewater treatment. It needs major revision before accepting for publication.

In introduction section lack of literature review is missing, authors should read below relevant reference and consider for citation.

Nanophotocatalysis and Environmental Applications, 139-169, 2020.

International Journal of Hydrogen Energy, doi.org/10.1016/j.ijhydene.2019.03.149. 2019

Nanophotocatalysis and Environmental Applications, 83-105, 2019

Polyhedron 120, 169-174, 2016.

Proper scale is missing in SEM images, better to provide details in the image.

The Conclusion section is too short, it should be written properly. And also mention like can be reused many times, it is better to mention how many times can be used.

English and grammatical errors should be rectified during the revision of the paper.

Reviewer: 3

Comments to the Author(s)

How to improve the economy and recycling degree of low concentration phosphorus wastewater treatment is one of the difficulties of municipal domestic wastewater treatment. This paper described the adsorption of low concentration phosphorus from simulated waste water source using bamboo-based nanocellulose fiber loaded with iron hydroxide ($\text{Fe}(\text{OH})_3/\text{CNFs}$). As an environmentally friendly adsorbent material, it has great potential for application. On the other hand, it can provide technical support for China's continuous improvement of sewage discharge standards, and it can also provide a feasible way for the recovery and utilization of phosphorus resources.

1. There are different kinds cellulose materials in nature, the authors should tell what is the advantage of bamboo compared with others.

2. Loading of metal/metal oxides to carrier material result in decreased surface area and porosity due to void volume blockages. However, what we observed here is a significant increase in surface area after loading of $\text{Fe}(\text{OH})_3$?

3. Figures 4 and 5 are similar in meaning and suggest integration.

4. Supplementary dynamics research to enhance the integrity of the article.

Author's Response to Decision Letter for (RSOS-190764.R0)

See Appendix A.

Decision letter (RSOS-190764.R1)

12-Aug-2019

Dear Professor Chen:

Title: Preparation and regeneration of iron-modified nanofibres for low concentration phosphorus-containing wastewater treatment
Manuscript ID: RSOS-190764.R1

It is a pleasure to accept your manuscript in its current form for publication in Royal Society Open Science. The chemistry content of Royal Society Open Science is published in collaboration with the Royal Society of Chemistry.

RSC Associate Editor
Comments to the Author:
(There are no comments.)

Reviewer(s)' Comments to Author:

Appendix A

List of Responses

Dear Editors and Reviewers:

First of all, we would like to thank the anonymous reviewers for their constructive comments to improve our manuscript entitled “ Preparation and regeneration of iron-modified nanofibres for low concentration phosphorus-containing wastewater treatment ” (ID: RSOS-190764). We really appreciate their precious time and great efforts on reviewing the manuscript. Consequently, we have studied comments carefully and have made correction which we hope meet with approval. Revised portion are marked in red in the paper. The main corrections in the paper and the responds to the reviewer’s comments are as flowing:

Responds to the reviewer’s comments:

Reviewer: 1

Comments to the Author(s)

Dear respected editor

Thank you very much for letting me the opportunity to reviews some scientific papers in your valuable journal.

Regarding the manuscript under consideration, (entitled: Preparation and regeneration of iron-modified nanofibres for low concentration phosphorus-containing wastewater

treatment), the manuscript and the obtained results are interesting and well organized.

I suggest this manuscript would be suitable for publication in your respected journal after minor revision. My comments are:

1. Following references must be included in the introduction

J. Inorg. Organomet. Polym. 29 (2019) 346–358

Compos. Part B. 166 (2019) 382–400.

Microchem. J. 145 (2019) 18-25.

J. Inorg. Organomet. Polym. 29 (2019) 229–247.

J. Mol. Liq. 253 (2018) 72–82.

J. Mol. Liq. 242 (2017) 364-374.

J. Mol. Liq. 248 (2017) 302–313.

J. Mol. Struct. 1143 (2017) 462–471.

J. Mol. Struct. 1086 (2015) 223-231.

J. Inorg. Organomet. Polym. (2017) 1-14.

Spectrochimica Acta Part A: Molecular and Biomolecular Spectroscopy
222(2019)117195

Adv. Powder Technol., 23 (2012) 757-760.

Response to comment:

We have already quoted in the introduction section.

2. There are some grammatical errors in the manuscript. Hence, the manuscript should

be carefully checked and necessary corrections should be done.

Response to comment:

We are very sorry for the grammatical mistakes in the article. According to the reviewer's opinion, we have revised the grammar of the whole article and revised portion are marked in red in the paper.

3. In the experimental part (3.1 Chemical reagents), the chemical structure of the used chemicals must be written.

Response to comment:

The 3.1 chemical reagents section has been modified and marked in red font. In detail:

The chemical reagents mainly included: sodium hydroxide (NaOH), potassium phosphate monobasic (KH₂PO₄), sulfuric acid (H₂SO₄), hydrogen chloride (HCl), ascorbic acid (C₆H₈O₆), ammonium molybdate ((NH₄)₆MO₇O₄·4H₂O), pot. antimony(iii)oxide tartrate hemi-hydrate (C₄H₄KO₇Sb·1/2H₂O), ferric nitrate (Fe(NO)₃·9H₂O), ferric chloride (FeCl₃·6H₂O), sodium chloride (NaCl), sodium nitrate (NaNO₃), sodium sulfate (Na₂SO₄), ammonium chloride (NH₄Cl), sodium carbonate (Na₂CO₃), potassium persulfate (K₂S₂O₈), ferric sulfate (Fe₂(SO₄)₃), and ferric reagent (LH-FE).

Reviewer: 2

Comments to the Author(s)

The manuscript reports on Preparation and regeneration of iron-modified nanofibres for low concentration phosphorus-containing wastewater treatment. Authors are attempted to develop iron-modified nanofibres for the treatment for low concentration phosphorus-containing wastewater treatment. It needs major revision before accepting for publication.

1. In introduction section lack of literature review is missing, authors should read below relevant reference and consider for citation.

Nanophotocatalysis and Environmental Applications, 139-169, 2020.

International Journal of Hydrogen Energy, doi.org/10.1016/j.ijhydene.2019.03.149.

2019

Nanophotocatalysis and Environmental Applications, 83-105, 2019

Polyhedron 120, 169-174, 2016.

Response to comment:

We have already quoted in the introduction section.

2. Proper scale is missing in SEM images, better to provide details in the image.

Response to comment:

We have modified the SEM images in the article.

3. The Conclusion section is too short, it should be written properly. And also mention

like can be reused many times, it is better to mention how many times can be used.

English and grammatical errors should be rectified during the revision of the paper.

Response to comment:

The 5. Conclusion section has been modified and marked in red font. In detail:

In this study, nanocellulose was produced using mechanical shearing of wet bamboo pulp as raw material. This method does not use TEMPO reagent, which simplifies the synthesis procedure and reduces secondary pollution from adsorbent production. After optimizing the preparation conditions of the iron-loaded nanocellulose, ferric chloride was determined to be the best modifier chemical. The specific surface area of the modified adsorbent increased from 14.48 m² g⁻¹ to 138.91 m² g⁻¹. When the mass ratio of Fe(OH)₃ to CNFs was 1.5:1, the addition of FeCl₃ solution and freeze-drying with liquid nitrogen yielded the highest adsorption capacity on phosphorus. The adsorption process of Fe(OH)₃@CNFs conforms to the pseudo-second-order kinetic model, suggesting that the adsorption process is chemical adsorption. After ten desorption-adsorption regeneration cycles, the phosphorus adsorption capacity remained stable. **The second generation iron-loaded nanocellulose can be reused at least ten times.** This iron-loaded nanocellulose holds great promise as highly effective and economically viable phosphorus adsorbent.

We are very sorry for the grammatical errors in the article. According to the reviewer's opinion, we have revised the grammar of the whole article and revised portion are marked in red in the paper.

Reviewer: 3

Comments to the Author(s)

How to improve the economy and recycling degree of low concentration phosphorus wastewater treatment is one of the difficulties of municipal domestic wastewater treatment. This paper described the adsorption of low concentration phosphorus from simulated waste water source using bamboo-based nanocellulose fiber loaded with iron hydroxide ($\text{Fe}(\text{OH})_3@ \text{CNFs}$). As an environmentally friendly adsorbent material, it has great potential for application. On the other hand, it can provide technical support for China's continuous improvement of sewage discharge standards, and it can also provide a feasible way for the recovery and utilization of phosphorus resources.

1. There are different kinds cellulose materials in nature, the authors should tell what is the advantage of bamboo compared with others.

Response to comment:

According to the reviewer's opinion, we have explained the advantages of bamboo in the third paragraph of the introduction. In detail:

Bamboo is a cellulose-rich material with a high cellulose content. China has a large bamboo production, and bamboo grows fast. The bamboo resources can be quickly regenerated. It is a high quality cellulose material. In recent years, due to the high surface area of nanomaterials, researchers are working on the preparation of nanomaterials as nanosorbents. And there are few studies on bamboo based cellulose at

the nanoscale. Therefore, it is valuable to study the application of bamboo nanofibers adsorbent.

2. Loading of metal/metal oxides to carrier material result in decreased surface area and porosity due to void volume blockages. However, what we observed here is a significant increase in surface area after loading of Fe(OH)₃ ?

Response to comment:

Fe(OH)₃ is loaded on the surface of the material and inside the pores after the material modification, the particles of ferric hydroxide may increase the specific surface area of the material. Has been explained in section 4.1.2.

3. Figures 4 and 5 are similar in meaning and suggest integration.

Response to comment:

According to the reviewer's opinion, we have already modified in the article.

4. Supplementary dynamics research to enhance the integrity of the article.

Response to comment:

We apologize for the loss of this important part of the dynamics. The related details about the adsorption kinetics was added in 4.4 section.